# Efficient Inference for Dynamic Topic Modeling with Large Vocabularies

**Federico Tomasi**[1]      **Mounia Lalmas**[1]      **Zhenwen Dai**[1]

[1]Spotify Research

## Abstract

Dynamic topic modeling is a well established tool for capturing the temporal dynamics of the topics of a corpus. In this work, we develop a scalable dynamic topic model by utilizing the correlation among the words in the vocabulary. By correlating previously independent temporal processes for words, our new model allows us to reliably estimate the topic representations containing less frequent words. We develop an amortised variational inference method with self-normalised importance sampling approximation to the word distribution that dramatically reduces the computational complexity and the number of variational parameters in order to handle large vocabularies. With extensive experiments on text datasets, we show that our method significantly outperforms the previous works by modeling word correlations, and it is able to handle real world data with a large vocabulary which could not be processed by previous continuous dynamic topic models.

## 1 INTRODUCTION

Topic modeling has been widely used to extract the main topics from a large collection of content such as text documents, images and other types of data that can be represented as bag-of-words [Balikas et al., 2016, Kho et al., 2017]. In topic modeling, a topic is represented as a probability distribution over the words in the vocabulary, and the words in a document are assumed to be independently drawn from a mixture of topics [Blei et al., 2003]. This approach allows us to efficiently infer topic compositions of documents in a large corpus without modeling sentence and paragraph structures.

Topic modeling has been extended to analyse the evolution of topics in a corpus over time [Bhadury et al., 2016,

Jähnichen et al., 2018, Tomasi et al., 2020]. The prior distribution of topic composition and the representations of individual topics are augmented into temporal processes such as Gaussian processes (GPs). With these methods, one can understand the rise and fall of a topic at an aggregated level. For example, when applied to the machine learning research literature, we can easily observe the changes of the popularity of different research topics over time. Such dynamic topic modeling requires a large amount of data because we need lots of documents at each time point to reliably estimate the topic representations and topic compositions. It is particularly challenging for modeling less frequent words because each word needs to be observed multiple times for each relevant topic at every time point, which is less likely for rare words. It is also computationally very challenging for the current dynamic topic models to handle a large vocabulary due to the increased computational and memory requirement. On the other hand, those less frequent words are often very specific, and hence may be strong cues to inform the topic of a document.

To leverage the information from less frequent words, we propose to incorporate word correlation in dynamic topic modeling. By doing so, a topic model can obtain sufficient signals about less frequent words by observing the existence of similar words. We incorporate word correlation by augmenting the generative model of topic representations. Previously, a topic representation is assumed to be drawn from a temporal process represented by a set of GPs, one for each word. The word correlation is introduced by correlating these previously independent GPs, resulting in a multi-output Gaussian process [MOGP; Álvarez et al., 2010]. Ideally, MOGP can explicitly capture word correlation in the form of a covariance matrix of all the words. This is infeasible due to both the high computational complexity and the large amount of data required for a reliable estimate. Instead of an explicit covariance matrix, we represent the correlation of words by embedding them into a latent space and generate the covariance matrix through a covariance function. With a Bayesian treatment to word representa-

*Accepted for the 38$^{th}$ Conference on Uncertainty in Artificial Intelligence* (UAI 2022).

tions (vectors) in the latent space, we can obtain a reliable estimate of the correlation with a small amount of data.

We develop an efficient stochastic variational inference method for the topic model. By extending the sparse GP [Titsias, 2009] to our MOGP formulation, the derived variational lower bound has the same computational complexity as in previous works with word independence and has significantly less number of variational parameters. Furthermore, compared to [Tomasi et al., 2020], we improve the amortised inference formulation by adopting a meta-encoder for the variational posterior of topic mixing proportions. The meta-encoder takes as inputs not only a document representation but also a summary of all the topic representations at a given time point. This allows the meta-encoder to easily handle the changes to topic representations. As the word distributions are modelled in as unnormalised log probability, the log-likelihood is calculated by drawing a sample of all the words in the vocabulary, which is computationally expensive for large vocabularies. For efficient inference, we derive an asymptotically unbiased estimator for the gradient of the lower bound that samples a subset of words from the vocabulary. This greatly increases the scalability of the inference method on large vocabularies. In summary, the main contributions of this paper are:

- We develop a word-correlated dynamic topic model, where the word correlation is jointly inferred together with the topic model from the data.

- We derive an efficient amortised variational inference method, which has the same computational complexity as the word independent model and less number of variational parameters.

- We derive an asymptotically unbiased estimator for the gradient of the lower bound, in which the computational complexity is constant with respect to the vocabulary size.

- On synthetic and real world datasets, we show that our method significantly outperforms the previous dynamic topic models in term of both quality and scalability.

**Outline.** The rest of this paper is organised as follows. Section 2 discusses related work. Section 3 presents our novel contribution, a word-correlated dynamic topic model. Section 4 describes an efficient variational inference procedure, using sparse Gaussian processes. Section 5 includes our experiments. Section 6 concludes with a discussion of the contributions of this work.

## 2 RELATED WORK

**Topic Models.** Topic models were proposed as a way to infer a mixture of topics from a collection of documents [Blei et al., 2003]. The correlated topic model [CTM; Blei and Lafferty, 2006a] allows topics to be correlated using a logistic normal distribution. Dynamic topic models have been proposed to enable consistent topics over a series of documents indexed by a temporal index [Blei and Lafferty, 2006b, Wang et al., 2008b, Dieng et al., 2019]. Recent models have been extending the idea by using the inherent structure between documents through continuous processes [Bhadury et al., 2016, Jähnichen et al., 2018, Tomasi et al., 2020]. However, the assumption of word independence given the topic does not allow information sharing across words, which limits in practice the applicability of topic models on corpus with large vocabulary and short documents.

**Word & Topic Embedding.** The idea of word or topic embedding has been explored in the topic modeling literature. In particular they have been used to learn coherent topics through word similarities [Xie et al., 2015] or represent a topic as Gaussian distributions in the embedded space [Xun et al., 2017]. Recently, topic modeling has been formulated as factor analysis, where words are embedded into a latent space [Yi et al., 2020]. Compared to our approach, none of these methods consider temporal dynamics and word embeddings are either learned outside the topic model such as using Word2Vec or lead to dramatic changes to the topic model formulation. Other recent work shows how a dynamic LDA and word embeddings can be effectively combined [Dieng et al., 2019]. Similar to our framework, the word embeddings are learned within the model and separated from the topic representations. However only discrete time stamps are considered, which do not allow to generalise the model to new time points, and topics are independent from each other. Another attempt to consider temporal dynamics have been proposed through Gaussian process latent variable models (GPLVMs), to infer a latent correlation between topics in discrete time stamps [Song et al., 2008]. Additional approaches consider embedding words through GPLVMs, and regard the resulting latent representation as topics. Topic correlation is then encoded through embedding similarity [Agovic and Banerjee, 2010, Hennig et al., 2012]. However, such topic models still use word embeddings to drive topic correlation. As we consider the correlation between the topics as an additional parameter, we can independently use (and learn from scratch) word embeddings to reliably model short texts.

**Multi-output Gaussian Process.** MOGPs [Álvarez et al., 2010, Williams et al., 2007, Stegle et al., 2011] extend GPs by explicitly modeling the correlation among multiple output dimensions. The correlation is encoded as a covariance matrix among output dimensions, which is also known as a coregionalization matrix. The resulting model is still a GP but with a much larger covariance matrix (a Kronecker product between the coregionalization matrix and the covariance matrix based on inputs), which poses a significant challenge on computation. Dai et al. [2017] addressed this problem

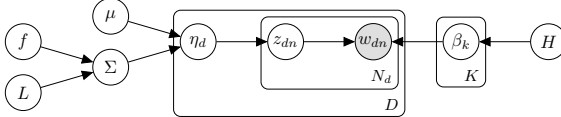

Figure 1: The graphical model for MIST.

by proposing an efficient variational inference method, in which the coregionalization matrix is represented as a latent space embedding similar to Gaussian process latent variable model [GPLVM; Lawrence, 2004]. Our approach extends the latent space embedding formulation of MOGP into topic modeling, which allows us to correlate the temporal processes of individual words in a topic representation.

**Adaptive Softmax.** The main issue in topic models is that the normalisation constant of the topic distribution depends on all of the vocabulary. The same issue can be found in language modeling or classification problems, where the number of classes to predict may be high, and computing probabilities for negative classes is too expensive [Bengio and Senécal, 2008, Blanc and Rendle, 2018]. Solutions to this problem have been proposed by approximating the softmax transformation [Zoph et al., 2016, Panos et al., 2021, Jean et al., 2014, Bamler and Mandt, 2020]. In particular, a self-normalised importance sampling procedure has been shown to effectively increase the training performance [Bengio and Senécal, 2008, Jean et al., 2014]. While not directly applicable in topic models, we show how it is possible to implement a similar procedure when estimating the topic distribution for dynamic topic models, allowing to effectively overcome vocabulary restrictions.

## 3 DYNAMIC TOPIC MODELING WITH WORD CORRELATION

We propose a dynamic topic model with word correlation, which we refer to as MIST (Multi-output with Importance Sampling Topic model). MIST is a probabilistic generative model that assumes that each document $d$, associated with a specific time point $\boldsymbol{x}_d$, is generated by sampling a set of words according to $K$ topics. Each document has an unnormalised topic mixing proportion $\boldsymbol{\eta}_d$ sampled from a prior distribution, $\boldsymbol{\eta}_d \sim \mathcal{N}(\boldsymbol{\mu}_{\boldsymbol{x}_d}, \Sigma_{\boldsymbol{x}_d})$, where $\boldsymbol{\mu}_{\boldsymbol{x}_d}$ is the mean of the distribution of topics mixing proportions associated to the time point $\boldsymbol{x}_d$, and $\Sigma_{\boldsymbol{x}_d}$ is the covariance matrix of topics at $\boldsymbol{x}_d$. When $\Sigma_{\boldsymbol{x}_d}$ is diagonal, the mixing proportion for each topic are independent to each other. Then, each word $w_n$ in this document is assigned with a topic $z_n$, which is sampled from the distribution $\sigma(\boldsymbol{\eta}_d)$, where $\sigma(x)_i = \exp(x_i)/\sum_j \exp(x_j)$ is the softmax function. Finally, the word $w_n$ is sampled by picking a word from the vocabulary following the unnormalised word distribution of the assigned topic $z_n$ at the time $\boldsymbol{x}_d$, $\boldsymbol{\beta}_{z_n}^{(\boldsymbol{x}_d)}$.

Figure 1 shows an overview of the graphical model of MIST. The generative process of a $N_d$-word document $d$ is summarised as follows. First, draw a mixture of topics $\boldsymbol{\eta}_d \sim \mathcal{N}(\boldsymbol{\mu}_{\boldsymbol{x}_d}, \Sigma_{\boldsymbol{x}_d})$. Then, for each word $n = 1, \ldots, N_d$:

1. Draw a topic assignment $z_n | \boldsymbol{\eta}_d$ from a categorical distribution with parameter $\sigma(\boldsymbol{\eta}_d)$;

2. Draw a word $w_n | z_n, \boldsymbol{\beta}$ from a categorical distribution with parameter $\sigma(\boldsymbol{\beta}_{z_n}^{(\boldsymbol{x}_d)})$.

The individual documents are assumed to be i.i.d. given the document-topic proportion and topic-word distribution. Under this generative process, the marginal likelihood for a given corpus $W$ that contains $D$ documents becomes:

$$p(W|\boldsymbol{\mu}, \Sigma, \boldsymbol{\beta}) =$$
$$\prod_{d=1}^{D} \int \prod_{n=1}^{N_d} \left( \sum_{z_n=1}^{K} p(w_{dn}|z_n, \boldsymbol{\beta}_{z_n}^{(\boldsymbol{x}_d)}) p(z_n|\boldsymbol{\eta}_d) \right) \quad (1)$$
$$p(\boldsymbol{\eta}_d | \boldsymbol{\mu}_{\boldsymbol{x}_d}, \Sigma_{\boldsymbol{x}_d}) \mathrm{d}\boldsymbol{\eta}_d.$$

To model the temporal dynamics of topic mixing proportions $\boldsymbol{\eta}_d$, we consider temporal processes as the prior distributions for $\mu$ and $\Sigma$. In particular, we consider zero-mean Gaussian process to model the topic probability $(\boldsymbol{\mu}_{\boldsymbol{x}_d})_{d=1}^{D}$, i.e., $p(\boldsymbol{\mu}) = \mathcal{GP}(\mathbf{0}, \kappa_\mu)$. Similarly, we model covariance matrices $(\Sigma_{\boldsymbol{x}_d})_{d=1}^{D}$ as generalised Wishart process (GWP), indicated as $\Sigma_{\boldsymbol{x}_d} \sim \mathcal{GWP}(V, \nu, \kappa_\theta)$ [Wilson and Ghahramani, 2011, Heaukulani and van der Wilk, 2019, Tomasi et al., 2020].

**Word Correlation.** In dynamic topic models [Jähnichen et al., 2018] the topic representations $\boldsymbol{\beta}$ are allowed to change over time. This is done by defining a GP prior over time independently for each word in each topic, so that there will be $KP$ independent GPs, where $P$ is the number of words in the vocabulary. This does not allow information sharing among similar words and results into a large number of variational parameters for inference. In this paper, we introduce correlation among words by defining a correlated temporal process for all words. First, we define a latent representation $\boldsymbol{h}_i \in \mathbb{R}^Q$ for each word in the vocabulary. The latent representations are given an uninformative prior $\boldsymbol{h}_i \sim \mathcal{N}(0, \mathbf{I})$. Then, a MOGP is defined for the topic representations over time for each topic:

$$p((\boldsymbol{\beta}_k)_{:}|H, \boldsymbol{x}) = \mathcal{N}((\boldsymbol{\beta}_k)_{:}|0, K^H \otimes K^{\boldsymbol{x}}), \quad (2)$$

where $(\cdot)_{:}$ denotes a matrix vectorisation, $\otimes$ denotes the Kronecker product, $\boldsymbol{\beta}_k$ is a $T \times P$ matrix representing the unnormalised word probabilities over time for the topic $k$ ($T$ is the number of unique time points in the corpus). The covariance matrix $K^{\boldsymbol{x}}$ is computed using the kernel function $\kappa_{\boldsymbol{x}}$ over all the time points $\boldsymbol{x}$ and the covariance matrix $K^H$ is computed using the kernel function $\kappa_H$ over all the word representations $H = (\boldsymbol{h}_1, \ldots, \boldsymbol{h}_P)$. With this

formulation, all the words at all the time points are jointly modeled with a single GP, in which the word correlation is encoded in the $TP \times TP$ covariance matrix. The prior distributions among different topics are assumed to be independent: $p(\boldsymbol{\beta}|\boldsymbol{x}, H) = \prod_{k=1}^{K} p(\boldsymbol{\beta}_k|\boldsymbol{x}, H)$. Note that the word correlations are encoded through the latent representations of words, which are static over time and shared across all the topics. Although the number of those latent vectors is relatively large, they can be reliably estimated by conditioning on the whole corpus.

The topic assignment variables $\{z_n\}_{n=1}^{N_d}$ for individual words of each document are latent and discrete, which are difficult to infer with variational inference. We marginalise out these discrete variables and obtain a closed form likelihood distribution,

$$p(W_d|\boldsymbol{\eta}_d, \boldsymbol{\beta}) = \prod_{n=1}^{N_d} \mathrm{Cat}(\sigma(\boldsymbol{\beta}^{(\boldsymbol{x}_d)}\sigma(\boldsymbol{\eta}_d))), \qquad (3)$$

where $\boldsymbol{\beta}^{(\boldsymbol{x}_d)}$ denotes the representations of all the topics at the time $\boldsymbol{x}_d$. With this formulation, a document can be represented in the form of word-count, allowing for a simplified formulation of our variational inference procedure.

# 4 VARIATIONAL INFERENCE

The MOGP formulation provides an elegant framework to correlate both the temporal dimension and the words in the vocabulary for the topic representations under a single GP. It also brings a significant challenge for inference, because the computational complexity for calculating the probability density function (PDF) of Equation (2) alone is $O(P^3 T^3)$. To overcome this challenge, we develop an efficient variational inference method based on the stochastic variational sparse GP formulation [SVGP; Hoffman et al., 2013], reducing the computational complexity to be linear with respect to $P$ and $T$.

## 4.1 VARIATIONAL INFERENCE FOR WORD CORRELATION

The word correlation is encoded by the latent representations of individual words $H$. We parameterise the variational posterior of $H$ as $q(H) = \mathcal{N}(m_H, S_H)$ and derive a variational lower bound,

$$\log p(W|\boldsymbol{x}) \ge \mathbb{E}_{q(H)}[\log p(W|\boldsymbol{x}, H)] - \mathrm{KL}(q(H)||p(H)), \qquad (4)$$

where $\mathrm{KL}(\cdot||\cdot)$ denotes the Kullback-Leibler divergence. The KL term in (4) can be computed in closed form because both $q(H)$ and $p(H)$ are normal distributions, but $p(W|\boldsymbol{x}, H)$ is intractable.

To derive a lower bound for the marginalised likelihood $p(W|\boldsymbol{x}, H)$, we first derive a variational lower bound for

$\log p(\boldsymbol{\beta}_k|\boldsymbol{x}, H)$ by taking the SVGP formulation. To take advantage of the Kronecker product structure in the covariance matrix, i.e., $K^H \otimes K^{\boldsymbol{x}}$, we define the inducing variables to be on a grid in the joint space of the word embedding and the temporal dimension. Let $U_{\boldsymbol{\beta}_k}$ be a $M_x \times M_H$ matrix, which follows the distribution $p(U_{\boldsymbol{\beta}_k}|Z_{\boldsymbol{x}}, Z_H) = \mathcal{N}((U_{\boldsymbol{\beta}_k}):|0, K_{uu})$, where $K_{uu} = K_{uu}^H \otimes K_{uu}^{\boldsymbol{x}}$. The rows of $U_{\boldsymbol{\beta}_k}$ corresponds to a set of inducing inputs in the temporal dimension, denoted as $Z_{\boldsymbol{x}}$, and the columns of $U_{\boldsymbol{\beta}_k}$ corresponds to a set of inducing inputs in the word embedding space, denoted as $Z_H$. Then, $K_{uu}^H$ is computed on the set of inducing inputs $Z_H$ with $\kappa_H$, while $K_{uu}^{\boldsymbol{x}}$ is computed on the set $Z_{\boldsymbol{x}}$ with $\kappa_{\boldsymbol{x}}$.

After defining the inducing variable $U_{\boldsymbol{\beta}_k}$, we reformulate $p(\boldsymbol{\beta}_k|\boldsymbol{x}, H)$ as

$$p(\boldsymbol{\beta}_k|\boldsymbol{x}, H) = \int p(\boldsymbol{\beta}|U_\beta, \boldsymbol{x}, H, Z_{\boldsymbol{x}}, Z_H) p(U_\beta|Z_{\boldsymbol{x}}, Z_H) \mathrm{d}U_\beta. \qquad (5)$$

The conditional distribution of $\boldsymbol{\beta}_k$ is

$$p(\boldsymbol{\beta}_k|U_\beta, Z_{\boldsymbol{x}}, Z_H, \boldsymbol{x}, H) = \qquad (6)$$
$$\mathcal{N}(\boldsymbol{\beta}_k|K_{fu}K_{uu}^{-1}U_\beta, K_{ff} - K_{fu}K_{uu}^{-1}K_{uf}), \qquad (7)$$

where $K_{fu} = K_{fu}^H \otimes K_{fu}^{\boldsymbol{x}}$ and $K_{ff} = K_{ff}^H \otimes K_{ff}^{\boldsymbol{x}}$. $K_{ff}^H$ is the covariance matrix computed on $H$ with $\kappa_H$, and $K_{ff}^{\boldsymbol{x}}$ is computed on $\boldsymbol{x}$ with $\kappa_{\boldsymbol{x}}$.

With the augmented GP formulation, we can derive a variational lower bound following [Hoffman et al., 2013]. However, a naive parameterisation of the variational posterior $q(U_{\boldsymbol{\beta}_k})$ using a multivariate normal distribution has a $M_x M_H \times M_x M_H$ covariance matrix, which is too large for matrix inversion. Instead, we define $q(U_{\boldsymbol{\beta}_k})$ with a Kronecker product covariance matrix similar to $p(U_{\boldsymbol{\beta}_k})$,

$$q(U_{\boldsymbol{\beta}_k}) = \mathcal{N}((U_{\boldsymbol{\beta}_k}):|M_:, \Sigma^H \otimes \Sigma^{\boldsymbol{x}}). \qquad (8)$$

where $M$ is the mean of the variational posterior, $\Sigma^H$ is a $P \times P$ covariance matrix and $\Sigma^{\boldsymbol{x}}$ is a $T \times T$ covariance matrix. With this formulation, the covariance matrix can be inverted efficiently by only inverting the two smaller covariance matrices, $(\Sigma^H \otimes \Sigma^{\boldsymbol{x}})^{-1} = (\Sigma^H)^{-1} \otimes (\Sigma^{\boldsymbol{x}})^{-1}$. This parameterisation also dramatically reduces the number of variational parameters in the covariance matrix from $M_x^2 M_H^2$ to $M_x^2 + M_H^2$.

With the variational posterior $q(U_{\boldsymbol{\beta}_k})$, we derive the variational lower bound for any downstream variable that consumes $\boldsymbol{\beta}_k$,

$$\log p(\cdot|H) \ge \mathbb{E}_{q(\boldsymbol{\beta}_k|H)}[\log p(\cdot|\boldsymbol{\beta}_k)] - \mathrm{KL}(q(U_{\boldsymbol{\beta}_k})||p(U_{\boldsymbol{\beta}_k})), \qquad (9)$$

where $q(\boldsymbol{\beta}_k|H) = \int p(\boldsymbol{\beta}_k|U_{\boldsymbol{\beta}_k}, H) q(U_{\boldsymbol{\beta}_k}) \mathrm{d}U_{\boldsymbol{\beta}_k}$. As the expectation $\mathbb{E}_{q(\boldsymbol{\beta}_k|H)}[p(\cdot|\boldsymbol{\beta}_k)]$ has no close form solution for

our model, we approximate it with Monte Carlo Integration by drawing samples from $q(\boldsymbol{\beta}_k|H)$.

The multivariate normal distribution with a Kronecker product covariance matrix like $p(U_{\boldsymbol{\beta}_k})$ and $q(U_{\boldsymbol{\beta}_k})$ is also called matrix normal distribution [Gupta and Nagar, 1999]. In matrix normal distribution notation, $q(U_{\boldsymbol{\beta}_k})$ becomes $\mathcal{MN}(M, \Sigma^H, \Sigma^{\boldsymbol{x}})$. Sampling from the distribution and the KL divergence can be computed efficiently (details in Supplementary Material).

**Sampling from** $q(\boldsymbol{\beta}_k|H)$. To compute the expectation in (9), we need to draw samples from $q(\boldsymbol{\beta}_k|H)$. As $q(\boldsymbol{\beta}_k|H)$ is a multivariate normal distribution with a full covariance matrix, drawing a correlated sample of $\boldsymbol{\beta}_k$ is computationally very expensive, $O(P^3T^3)$. Usually, we can avoid drawing a fully correlated sample if $\boldsymbol{\beta}_k$ in the downstream log PDF, $\log p(\cdot|\boldsymbol{\beta}_k)$, can be decomposed into a sum of individual entries, *e.g.*, $p(\cdot|\boldsymbol{\beta}_k)$ is a normal distribution. However, due to the softmax function that is applied to $\boldsymbol{\beta}_k$ in (1), such decomposition is not applicable to our model. To efficiently sample from $q(\boldsymbol{\beta}_k|H)$, we apply another sparse GP approximation, the FITC approximation [Naish-Guzman and Holden, 2008], to the conditional distribution of $\boldsymbol{\beta}_k$. The resulting formulation is

$$
\begin{aligned}
p_{\text{FITC}}(\boldsymbol{\beta}_k|U, Z_{\boldsymbol{x}}, Z_H, \boldsymbol{x}, H) = \\
\mathcal{N}(\boldsymbol{\beta}_k|K_{fu}K_{uu}^{-1}(U^\top)_{:}, \text{diag}\left(K_{ff} - K_{fu}K_{uu}^{-1}K_{uf}\right)),
\end{aligned}
\tag{10}
$$

where diag $(\cdot)$ returns a diagonal matrix while keeping the diagonal entries. Since $K_{fu}$, $K_{ff}$ and $K_{uu}$ have a Kronecker structure, we can rewrite mean and covariance to compute them efficiently. Sampling from (10) is efficient because individual entries of $\boldsymbol{\beta}_k$ can be sampled independently. This reduces the computational complexity of sampling $\boldsymbol{\beta}_k$ from $O(P^3T^3)$ to $O(PTM_x^2M_H^2)$.

## 4.2 VARIATIONAL INFERENCE FOR MIXTURE OF TOPICS

With a variational posterior $q(\boldsymbol{\eta}_d)$ for each document, we can derive a variational lower bound of the log probability over the documents as:

$$
\begin{aligned}
\log p(W|\boldsymbol{\mu}, \Sigma, \boldsymbol{\beta}) \geq \\
\sum_{d=1}^{D} \Big( \mathbb{E}_{q(\boldsymbol{\beta}|H)q(\boldsymbol{\eta}_d)} \left[ \log p(W_d|\boldsymbol{\eta}_d, \boldsymbol{\beta}^{(\boldsymbol{x}_d)}) \right] \\
- \text{KL}\left( q(\boldsymbol{\eta}_d)||p(\boldsymbol{\eta}_d|\boldsymbol{\mu}_{\boldsymbol{x}_d}, \Sigma_{\boldsymbol{x}_d}) \right) \Big) \\
= \mathcal{L}_W.
\end{aligned}
\tag{11}
$$

Since the lower bound is a summation over individual documents, this formulation allows for a stochastic approximation by sub-sampling the documents.

**Importance Sampling.** Computing the expectation $\mathbb{E}_{q(\boldsymbol{\beta}|H)q(\boldsymbol{\eta}_d)} \left[ \log p(W_d|\boldsymbol{\eta}_d, \boldsymbol{\beta}^{(\boldsymbol{x}_d)}) \right]$ is still problematic when the number of words in the vocabulary increase, as we need to sample each word to compute the normalisation constant the softmax function as in Equation (3). First, let $\boldsymbol{\xi}_d = \boldsymbol{\beta}^{(\boldsymbol{x}_d)}\sigma(\boldsymbol{\eta}_d)$, and $\boldsymbol{\xi}_{d,n} = (\boldsymbol{\xi}_d)_n$. We can rewrite (3) as $p(W_d|\boldsymbol{\eta}_d, \boldsymbol{\beta}) = \prod_{n=1}^{N_d} \text{Cat}(\sigma(\boldsymbol{\xi}_d)) = \tilde{\mathcal{L}}_W$. Then, we can explicitly write its derivative as (details in Supplementary Material):

$$
\nabla\tilde{\mathcal{L}}_W =
$$
$$
\mathbb{E}_{q(\boldsymbol{\beta}|H)q(\boldsymbol{\eta}_d)} \sum_{n=1}^{N_d} \left[ \nabla\boldsymbol{\xi}_{d,n} - \sum_{i=1}^{P} \frac{\exp(\boldsymbol{\xi}_d)_i}{\sum_{j=1}^{P} \exp(\boldsymbol{\xi}_d)_j}\nabla\boldsymbol{\xi}_{d,i} \right].
\tag{12}
$$

In the sum inside the parenthesis, it is clear we need to sample from all of the vocabulary (that has size $P$). This is inefficient and may even be unfeasible for a large vocabulary. The key idea to solve this problem, and efficiently scale our topic model to an arbitrary large set of words in the vocabulary, is to approximate the normalisation constant with a fixed number of words, using a self-normalising importance sampling [Bengio and Senécal, 2008]. Let consider the words appearing in the batch of documents under analysis as *positive* (*e.g.*, as in a positive class in a classification problem). We then borrow from the "extreme" classification literature the idea to use importance sampling to approximate the normalisation constant, which consists in consider a random sample of $M$ classes (in our case, words from the vocabulary) and using those to approximate the normalisation constant [Bamler and Mandt, 2020].

Consider a sample vector $\boldsymbol{s} \in \{1, ..., P\}^{M+N_d}$, which represents a sample of words in the vocabulary and stores the index of the $N_d$ positive (words appearing in document $d$) and the index of the $M$ sampled words. Let $\xi'_{d,i} := \xi_{d,i} - \ln(Q_{di}/P)$ if $y_i = 0$ (*i.e.*, word $i$ does not appear in document $d$), $\xi'_{d,i} := \xi_{d,i} - \ln(Q_{di})$ otherwise, with $Q_{di}$ proposal distribution. We shift the true logits by the expected number of occurrences of a word $i$, ensuring that the sampled softmax is asymptotically unbiased. In our experiment we choose $Q$ to be a uniform distribution over the subset of words considered, so $Q_{di} = 1/(N_d + M)$ [Jean et al., 2014]. Then:

$$
\begin{aligned}
\nabla\tilde{\mathcal{L}}_W \approx \mathbb{E}_{q(\boldsymbol{\beta}|H)q(\boldsymbol{\eta}_d)} \sum_{n=1}^{N_d} \bigg[ \nabla\boldsymbol{\xi}_{d,n} \\
- \sum_{i=1}^{M+N_d} \frac{\exp(\xi'_i)}{\sum_{j=1}^{M+N_d} \exp(\xi'_j)}\nabla\xi_{d,i} \bigg].
\end{aligned}
\tag{13}
$$

In this way, we further reduce the complexity of computing expectation from $O(PTM_x^2M_H^2)$ to $O((M + N_d)TM_x^2M_H^2)$.

**Documents meta-encoder.** We parameterise the variational posteriors $q(\boldsymbol{\eta}_d)$ for individual documents as:

$$q(\boldsymbol{\eta}_d) = \mathcal{N}(\phi_m([W_d \, M_{\beta,\boldsymbol{x}_d}]), \phi_S([W_d \, M_{\beta,\boldsymbol{x}_d}])), \quad (14)$$

where $\phi_m$ and $\phi_S$ are parametric functions generating the mean and variance of $q(\boldsymbol{\eta}_d)$, respectively, $M_{\beta,\boldsymbol{x}_d}$ is the mean of the GP prediction at the inducing point location $Z_H$, and $[A \, B]$ denotes the concatenation of the matrices $A$ and $B$. Instead of implicitly learning the topic information into $\phi_m$ and $\phi_S$ as in [Tomasi et al., 2020], we explicitly pass in a summary of all the topic representation at the time point $\boldsymbol{x}_d$. We define $M_{\beta,\boldsymbol{x}_d}$ as the mean of the GP prediction at inducing point location $Z_H$, to keep the complexity constant with respect to the number of words in the vocabulary. We treat such prediction as the summary of all the topic representations at input $\boldsymbol{x}_d$ because the inducing variable in sparse GP can be viewed as a summary of all the data [Titsias, 2009]. By having the topic representations as inputs, the encoder does not need to "memorise" the information about topics but rather link a document to relevant topic representations. Therefore we refer to $\phi_m$ and $\phi_S$ as the meta-encoder.

**Lower bound.** Note that the lower bound $\mathcal{L}_W$ is intractable. We compute an unbiased estimate of $\mathcal{L}_W$ via Monte Carlo sampling. As $q(\boldsymbol{\eta}_d)$ are normal distributions, we obtain a low-variance estimate of the gradients via the reparameterisation strategy [Kingma and Welling, 2014].

The document-topic proportion for each document $d$ follows a prior distribution $p(\boldsymbol{\eta}_d | \boldsymbol{\mu}_{\boldsymbol{x}_d}, \Sigma_{\boldsymbol{x}_d})$, where the Gaussian process $p(\boldsymbol{\mu}) = \mathcal{GP}(0, \kappa_\mu)$ provides the mean and the Wishart process $p(\Sigma) = \mathcal{GWP}(V, \nu, \kappa_\theta)$ provides the covariance matrix at $\boldsymbol{x}_d$. To enable efficient inference for both GP and GWP, we take a SVGP approach to construct the variational lower bound of our model [Tomasi et al., 2020]. We can derive the complete variational lower bound $\mathcal{L}$ of MIST combining the lower bounds (11), (4), (9) and the lower bounds for GPs and GWP (details in Supplementary Material).

## 5 EXPERIMENTS

We compared MIST with several static and dynamic topic models: *(i)* LDA with an mean-field variational inference [Hoffman et al., 2010, Pedregosa et al., 2011]; *(ii)* CTM with variational inference [Blei and Lafferty, 2006a];[1]; *(iii)* dynamic embedded topic model [DETM; Dieng et al., 2019], and *(iv)* dynamic correlated topic model [DCTM; Tomasi et al., 2020]. We do not compare with other previously proposed dynamic models that only cater for independent topics, *e.g.*, DTM [Blei and Lafferty, 2006b], FastDTM [Bhadury et al., 2016] (do not handle continuous dynamics),

---

[1]We additionally infer the variational posteriors for the topic representations $\boldsymbol{\beta}$, of which the point estimates are inferred in [Blei and Lafferty, 2006a].

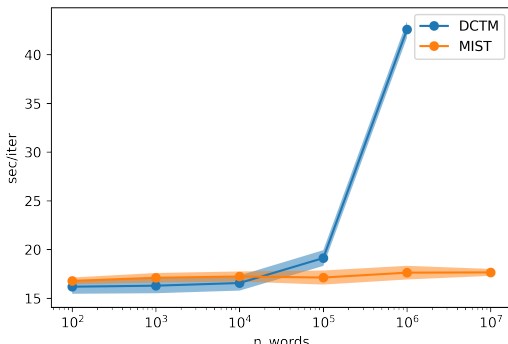

Figure 2: Average time to compute 5 epochs in a dataset with increasing number of words.

and gDTM [Jähnichen et al., 2018] (only considers dynamics for $\boldsymbol{\beta}$), as both DETM and DCTM have been shown to generalise and improve on such models.

**Performance Analysis.** We first empirically evaluated the benefit of MIST using synthetic datasets. We show the performance of MIST against the DCTM model [Tomasi et al., 2020] in Figure 2 on a dataset with increasing words (from 100 to 10M), while keeping a fixed number of samples at different time points. Our proposed model consistently outperforms DCTM across all the vocabulary sizes. In particular, the computational benefit of MIST is more evident after reaching 100K (and 1M) words. On the dataset with 10M words we showed how DCTM could not be used anymore, as it was computationally intractable. Instead, we notice how MIST is able to scale to 10M words, with an average computational time that is lower than the time required by DCTM with 1M words. Similarly, MIST with 1M words took less than DCTM with 100K words.

**Quantitive Analysis.** We here showcase the benefit of incorporating word correlation in topic modelling by comparing MIST with state-of-the-art topic models on public datasets. The common parameters across the models have been kept the same for a fair comparison (*e.g.*, the number of topics). Here, we fix the number of topics to be small (30 topics for all datasets apart for SotU using 20) so we are able to compare to the baselines. In addition, our experiments do not show relevant differences across models when varying the number of topics (analysis in the Supplementary Material). In all datasets, there is a timestamp associated with each document. Static topic models (LDA and CTM) are optimised without considering the timestamps, while DCTM and MIST incorporate the continuous timestamps into the inference. For DETM, which considers discrete times, we discretised the timestamps into 30 bins to make the inference computationally feasible on our machine.

We split each dataset considering 75% of the samples as

Table 1: Average per-word perplexity (the lower the better) on public datasets.

| Dataset | #words | LDA | CTM | DETM | DCTM | MIST |
|---|---|---|---|---|---|---|
| Blogs | 3000 | 1538.74 | 1525.01 | 1305.34 | 1013.71 | **949.76** |
| SotU | 4583 | 3090.52 | 1937.19 | 3069.54 | 2205.01 | **1675.32** |
| NeurIPS | 4799 | 1321.51 | 1241.72 | 941.91 | 1012.51 | **888.59** |
| DoJ | 9591 | 1459.10 | 931.23 | 936.00 | 928.08 | **613.37** |
| Abstracts | 13126 | 3206.09 | 2918.36 | 2566.50 | - | **1857.67** |
| News | 22459 | 5351.83 | 3553.73 | 1957.25 | - | **1703.77** |
| Twitter | 83582 | 7101.83 | 8576.97 | 2721.05 | - | **2595.14** |

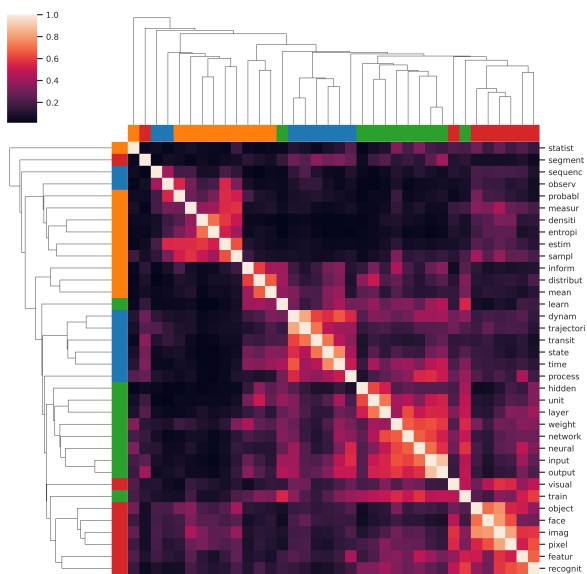

Figure 3: Correlation matrix for top-10 frequent words in four topics inferred by MIST for NeurIPS dataset.

training and 25% as test. Documents associated with the same time stamps were assigned to the same split. For each dynamic topic model we used a Matérn 3/2 kernel for $\beta$, to allow topics to quickly incorporate new words. This is important to incorporate neologisms, particularly for datasets such as NeurIPS conference papers and Elsevier (Abstracts) corpus, where the names of novel models become quoted in citations (for example, "LDA" starting to appear in publications together as "topic modeling" after its introduction in 2003). For the other parameters $\mu$ and $f$ we use a squared exponential kernel, as we expect a smooth temporal evolution of both topic probabilities and their correlation. Full details on data, model parameters and experimental settings can be found in the Supplementary Material.

We report the average per-word perplexity computed on the held-out test set of the datasets for all the models in Table 1. The per-word perplexity is a measure of best fit to compare models, computed as the exponential average negative predictive log-likelihood for each word [Wang et al., 2008a]:

$$\text{perpl}_{\text{pw}}(t) = \exp \left\{ -\frac{1}{|D_t|} \sum_{d \in D_+} \frac{\log p(W_d | \boldsymbol{\eta}_d, \boldsymbol{\beta}^{(\boldsymbol{x}_d)})}{N_d} \right\},$$

where $\log p(W_d)$ is estimated as in Equation (11). MIST consistently outperforms all the baselines on all the datasets. The benefit in using our models is more evident for the datasets that have a larger time span (SotU) or a shorter document size (Blogs, News, Twitter). There is also a significant performance gap between static and dynamic topic models, which demonstrates the advantages of incorporating temporal information into topic modeling. Comparing MIST to DCTM, the perplexity decreases on average by 10%, which shows that incorporating word correlation into topic modeling can significantly improve the quality of modeling. When using large scale datasets (more than 10K words) we were not able to run DCTM as the size of the vocabulary was intractable. However, we were able to use our model MIST, and obtain better perplexity than static topic models and the discrete time topic model DETM (for which we had to discretise the time stamps of the documents into 30 bins).

We note that dynamic topic models allows to keep the number of topics low with respect to static topic models, because

topics will be linked through time. Hence, if static models need a lot of topics to be able to cluster words in different time stamps, dynamic models allow to adapt the words related to the same topics to achieve the same (or best) results on a small number of topics. Furthermore, our proposed model, through the use of the Kronecker decomposition, is able to keep the number of parameters low in terms of words representation. Instead of independently learn the representation of words, we learn a representation of a similar group of words, which allows us to scale the model and train in the presence of few words in documents.

**Qualitative Analysis on NeurIPS dataset.** To provide insights about the word correlation in MIST, we visualise the inferred word correlation. We choose four interred popular topics across all years on the NeurIPS dataset and collect the top-10 frequent words for each topic (Supplementary Material). Then, we compute the covariance matrix among these frequent words (duplicate words are removed) by applying the learnt kernel function $\kappa_H$ to the mean of the variational posterior of the word representations $m_H$. The covariance matrix is converted into a correlation matrix for better interpretability. We visualise the resulting correlation matrix using a heatmap (Figure 3). The color map is shown in the left top corner of the plot. A lighter color indicates a stronger correlation and a darker color indicates a weaker correlation. Due to the choice of the kernel function (squared exponential) no anti-correlation is captured in the correlation matrix. We also applied a simple hierarchical clustering to the correlation matrix. With only the word relation, the words associated with the same topic are roughly grouped together (topics are here unknown to the clustering algorithm). For example, *network*, *weight*, *neural* and *layer*,

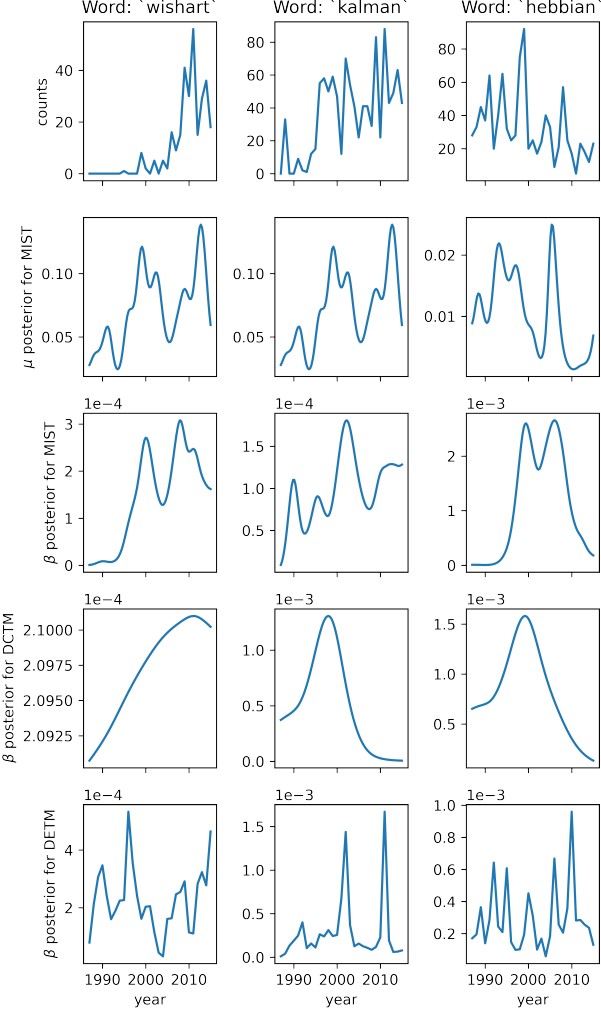

Figure 4: Word counts in dataset and estimated word probability in dynamic topic models for few low frequency words.

models we tested (we omit static topic models as the result would be a flat line of the word probability as it is independent of time). MIST is shown to be the most accurate in modelling these words, by capturing the general dynamics of the word in the dataset but without overfitting it.

For example, consider the word "Wishart" corresponding to the Wishart distribution (and process). The word counts are very low (less than 50 counts in total each year). However, the word is very much indicative of Bayesian inference topic, as the Wishart distribution is the conjugate prior of the inverse covariance-matrix of a multivariate-normal. Indeed, it has a high correlation with the much more common word "posterior" ($\rho = 0.688$ using the learnt $\kappa_H$ from MIST). MIST is able to accurately model the increasing dynamics of such word over time (middle row). Conversely, DCTM and DETM that considers words independently do not have enough data to accurately model its dynamics.

## 6 CONCLUSION

We developed an efficient approach to model word correlation in dynamic topic modeling. Our approach incorporates word dynamics through the use of multi-output Gaussian processes. We improved the amortised inference by proposing a meta-encoder, which allows MIST to be more sensitive to the changes of topic representations. Finally, we enable scalable inference for large vocabularies by deriving an asymptotically unbiased estimator of the gradient that allows us to dramatically subsample the number of words in computation. As shown in our experiments, incorporating word correlation into dynamic topic modeling significantly improves the modeling quality and allows topic models to leverage information from related words.

which identify the topic *neural network*, have a very similar embedding. The word pairs that are often used together in some research area show interesting strong correlations such as *input-output*, *imag-pixel*, *time-state*. This indicates that the word correlation has contributed to the identification of these topics.

We analyse the performance of MIST in terms of modeling infrequent keywords compared to previous methods (see Figure 4). We plot the per-year word counts of three keywords ("Wishart", "Kalman" and "Hebbian") in the Neurips datasets in the top row. All three words on averages appear less than 100 times a year, which are infrequent, but they are all clear indicators of ML and neuroscience sub-fields. We then compare the word probabilities of each word in the topic with the strongest connection to the word inferred by MIST, DCTM and DETM. We plot the posterior of $\mu$ in the topic inferred by MIST where the word is most prominent, and the posterior of $\beta$ for each one of the dynamic topic

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
