# OpenReview forum: "Efficient Inference for Dynamic Topic Modeling with Large Vocabularies"
_auai.org/UAI/2022/Conference — UAI 2022 Poster_

### Official Review · Reviewer_A8dE · 2022-04-12

**Q2(1) Originality/Novelty:** 2
**Q2(2) Significance/Impact:** 2
**Q2(3) Correctness/Technical Quality:** 2
**Q2(6) Clarity Of Writing:** 2
**Q6 Overall Score:** 5
**Q8 Confidence In Your Score:** 2

**Q1 Summary And Contributions:**

The authors develop a scalable dynamic topic model by utilizing the correlation among the words in the vocabulary. Specifically speaking, multi-output Gaussian processes are adopted instead of Gaussian processes. For model inference, the authors develop an amortised variational inference method with self-normalised importance sampling approximation to the word distribution that dramatically reduces the computational complexity.

**Q2 Assessment Of The Paper:**

More detailed information regarding each of these aspects is given below:

**Q2(4) Quality Of Experiments (Optional):**

2: Fair: The experimental evaluation is weak: important baselines are missing, or the results do not adequately support the main claims.

**Q2(5) Reproducibility:**

2: Fair: Key resources (e.g., proofs, code, data) are unavailable but key details (e.g., proof sketches, experimental setup) are sufficiently well-described for an expert to confidently reproduce the main results.

**Q3 Main Strengths:**

The motivation is clear and strong. A scalable dynamic topic model is proposed by utilizing the correlation among the words in the vocabulary. Specifically speaking, multi-output Gaussian processes are adopted instead of Gaussian processes. For model inference, the authors develop an amortised variational inference method with self-normalised importance sampling approximation to the word distribution that dramatically reduces the computational complexity.

**Q4 Main Weakness:**

It will be better to provide more evidence on scalability of proposed method from both the theoritical aspect and empirical aspect, e.g. computational complexity of the whole algorithm and more experiments to verify the scalablity of proposed method besides Figure 2.

**Q5 Detailed Comments To The Authors:**

It will be better to provide more evidence on scalability of proposed method from both the theoritical aspect and empirical aspect, e.g. computational complexity of the whole algorithm and more experiments to verify the scalablity of proposed method besides Figure 2.

**Q7 Justification For Your Score:**

The motivation is clear and the paper is well-organized. But I have some concern about the novelty and theoritical contribution of this paper.

**Q9 Complying With Reviewing Instructions:**

1: Yes.

---

### Official Review · Reviewer_nkBT · 2022-04-12

**Q2(1) Originality/Novelty:** 3
**Q2(2) Significance/Impact:** 3
**Q2(3) Correctness/Technical Quality:** 3
**Q2(6) Clarity Of Writing:** 3
**Q6 Overall Score:** 7
**Q8 Confidence In Your Score:** 3

**Q1 Summary And Contributions:**

The paper proposes a dynamic topic model with word correlation working particularly well with infrequent yet relevant words.

**Q2 Assessment Of The Paper:**

More detailed information regarding each of these aspects is given below:

**Q2(4) Quality Of Experiments (Optional):**

3: Good: The experimental evaluation is adequate, and the results convincingly support the main claims.

**Q2(5) Reproducibility:**

3: Good: Key resources (e.g., proofs, code, data) are available and key details (e.g., proofs, experimental setup) are sufficiently well-described for competent researchers to confidently reproduce the main results.

**Q3 Main Strengths:**

- the topic is relevant
- the relevant literature is handled well
– the evaluation is thorough for a conference paper

**Q4 Main Weakness:**

- boldness in claims might turn out as a liability in scientific papers, yet some more emphasis on some innovative aspects could have reduced the "flavour" of "incrementality" that is basically the only apparent flaw of the paper

**Q5 Detailed Comments To The Authors:**

Abstract is too long: it is half an intro; if you really focus on your contribution, you might help the reader actually appreciating the value of your work Conclusion, on the other hand, is a good example of synthesis


**Q7 Justification For Your Score:**

The paper is good. Still, it might be somehow incremental w.r.t. current state of the art: some rewriting by the authors might help addressing that,

**Q9 Complying With Reviewing Instructions:**

1: Yes.

---

### Official Review · Reviewer_4hW3 · 2022-04-13

**Q2(1) Originality/Novelty:** 3
**Q2(2) Significance/Impact:** 2
**Q2(3) Correctness/Technical Quality:** 3
**Q2(6) Clarity Of Writing:** 3
**Q6 Overall Score:** 7
**Q8 Confidence In Your Score:** 3

**Q1 Summary And Contributions:**

This paper proposed an efficient dynamic topic model which can leverage information from less frequent words by considering word correlations. Specifically, the authors utilize the multi-output Gaussian process (MOGP) to capture word correlations, which are generated from latent word embeddings.  The proposed model demonstrated higher efficiency and better performance compared with existing dynamics topic models on both synthetic and real datasets.

**Q2 Assessment Of The Paper:**

More detailed information regarding each of these aspects is given below:

**Q2(4) Quality Of Experiments (Optional):**

3: Good: The experimental evaluation is adequate, and the results convincingly support the main claims.

**Q2(5) Reproducibility:**

2: Fair: Key resources (e.g., proofs, code, data) are unavailable but key details (e.g., proof sketches, experimental setup) are sufficiently well-described for an expert to confidently reproduce the main results.

**Q3 Main Strengths:**

1. This paper is well-motivated. Modeling less frequent words is indeed a challenge for dynamic topic models and introducing word correlations to leverage those less frequent words is an interesting solution.
2. The proposed model is technically solid. The model takes word correlations into consideration without introducing extra computational complexity.
3. The evaluation part is very comprehensive. Especially the case studies on NeurIPS dataset intuitively show that the proposed model can better capture dynamics of less frequent words.


**Q4 Main Weakness:**

Some of the experiment settings and results need to be better explained.

**Q5 Detailed Comments To The Authors:**

1. The results on several public datasets (Table I) differs from the results reported in a related work [Tomasi et al., 2020] a lot. What is the reason of this difference?
2. The related work [Dieng et al., 2020] also utilized word embeddings for modeling word correlations. Why it is not compared as a baseline?


**Q7 Justification For Your Score:**

This paper is well-motivated. The method proposed in this paper is technically sound and achieved good performance with high efficiency. Therefore, I recommend accepting this paper.

**Q9 Complying With Reviewing Instructions:**

1: Yes.

---

### Official Review · Reviewer_KTc2 · 2022-04-20

**Q2(1) Originality/Novelty:** 2
**Q2(2) Significance/Impact:** 2
**Q2(3) Correctness/Technical Quality:** 3
**Q2(6) Clarity Of Writing:** 4
**Q6 Overall Score:** 3
**Q8 Confidence In Your Score:** 4

**Q1 Summary And Contributions:**

The paper proposes to jointly learn word-correlation along with dynamic topics. The further propose an efficient amortized inference method which has the same computational complexity as the word independent model. They also show that their proposed model outperforms previous dynamic topics models.


**Q2 Assessment Of The Paper:**

More detailed information regarding each of these aspects is given below:

**Q2(4) Quality Of Experiments (Optional):**

1: Poor: The experimental evaluation is flawed or the results fail to adequately support the main claims.

**Q2(5) Reproducibility:**

4: Excellent: Key resources (e.g., proofs, code, data) are available and key details (e.g., proof sketches, experimental setup) are comprehensively described for competent researchers to confidently and easily reproduce the main results.

**Q3 Main Strengths:**

The main proposal of the paper to develop a word correlated topic model is interesting.
The paper is well written and easy to follow.


**Q4 Main Weakness:**

There have been several efforts on capturing the correlation between words and this paper doesn’t do enough to differentiate itself from them. An example paper whose citation is missing is the Latent LSTM allocation paper and several papers that came after it. They capture dynamic relationships and are vastly scalable. In the absence of a comparison with some of the current state-of-the-art it is very difficult to clearly evaluate the paper and recommend it for acceptance.

Another thing that can improve the strength of the paper would be to add a use case where finding the word correlation is useful, perhaps in a downstream task.


**Q5 Detailed Comments To The Authors:**

Please see my comments in the weakness section.


**Q7 Justification For Your Score:**

The paper lacks novelty, and proper comparison to current baselines. They also need to show the applicability of the proposed model to a downstream task.


**Q9 Complying With Reviewing Instructions:**

1: Yes.

---

### Decision · Program_Chairs · 2022-05-15

**Decision:**

Accept (Poster)

**Comment:**

Meta Review: Most reviewers appreciated the contributions of the paper, the motivations, and the thoroughness of the experiments. There were some concerns raised about related work, which we hope that authors will address in the future.